# Design issues and solutions for stop-signal data from the Adolescent Brain Cognitive Development (ABCD) study

Patrick G Bissett*, McKenzie P Hagen, Henry M Jones, Russell A Poldrack

Department of Psychology, Stanford University, Stanford, United States

**Abstract** The Adolescent Brain Cognitive Development (ABCD) study is an unprecedented longitudinal neuroimaging sample that tracks the brain development of over 9–10 year olds through adolescence. At the core of this study are the three tasks that are completed repeatedly within the MRI scanner, one of which is the stop-signal task. In analyzing the available stopping experimental code and data, we identified a set of design issues that we believe significantly compromise its value. These issues include but are not limited to variable stimulus durations that violate basic assumptions of dominant stopping models, trials in which stimuli are incorrectly not presented, and faulty stop-signal delays. We present eight issues, show their effect on the existing ABCD data, suggest prospective solutions including task changes for future data collection and preliminary computational models, and suggest retrospective solutions for data users who wish to make the most of the existing data.

## Introduction

The Adolescent Brain Cognitive Development (ABCD) is the largest and most comprehensive long-term study of brain development and child health in the United States (*Casey et al., 2018*). The study includes 11,878 youth and their families and aims to understand the environmental, social, genetic, and other biological factors that affect brain and cognitive development. This study was made possible by the Collaborative Research on Addiction at NIH (CRAN) including the National Institute of Drug Abuse, National Institute on Alcohol Abuse and Alcoholism, and National Cancer Institute in partnership with the Eunice Kennedy Shriver National Institute of Child Health and Human Development, National Institute of Mental Health, National Institute of Minority Health and Health Disparities, National Institute of Neurological Disorders and Stroke, and the NIH Office of Behavioral and Social Sciences Research. CRAN granted $590 million in support to 21 research institutions across the United States to complete this study.

At the core of the ABCD study are the structural and functional MRI brain scans that occur biennially for each participant. The initial baseline scans have all been completed. The study organizers chose to include three cognitive tasks to be presented during fMRI acquisition: the monetary incentive delay task (*Knutson et al., 2000*), the emotional N-back task (*Barch et al., 2013*), and the stop-signal task (*Logan and Cowan, 1984*). In this manuscript, we focus solely on the stop-signal task.

We analyzed behavioral data from the baseline scan of 8,464 of the 11,878 participants. This subset resulted from the following exclusions. First, we attempted to download 8,811 participants from the 'FastTrack Recommended Active Series' from the NIMH Data Archive, which has gone through some quality assurance for the associated imaging files by the ABCD study organizers. Of these, 8,808 files were successfully downloaded. Of these, 8,734 files contained stop-signal task data that were encoded and formatted in text or csv format with header columns and each row representing one task trial. Finally, 270 subjects were removed who did not have two complete runs with 180 trials each, leaving us with a total of 8,464 complete datasets. We did not apply any other performance-

*For correspondence:
pbissett@stanford.edu

**Competing interests:** The authors declare that no competing interests exist.

based exclusion, including the ABCD study's performance flag (*Garavan et al., 2020*), as this would have obscured our ability to evaluate potential issues (e.g., it would have masked Issue 3, see below).

The stop-signal task is a primary paradigm used to understand response inhibition. It involves making a choice response to a go stimulus but attempting to stop that response when an infrequent stop signal occurs after a stop-signal delay (SSD). The dominant theoretical framework for understanding and interpreting stopping tasks is the independent race model (*Logan et al., 2014*; *Logan and Cowan, 1984*), which assumes that a go process begins when a go stimulus occurs and races independently against a stop process that begins when the stop stimulus occurs. The go process finishing first results in a stop failure (an overt response), whereas the stop process finishing first results in stop success (no response).

The ABCD study is laudable for many reasons, not least of which is the dedication to making their data and materials openly available. This includes but is not limited to experimental code, trial-by-trial behavioral performance in each task, functional neuroimaging data, and structural neuroimaging data. Our research group aimed to use these data to understand the behavioral and neural underpinnings of response inhibition. In doing so, we first evaluated the experimental code and behavioral performance in the stop-signal task.

During our analyses, we found a nexus of issues with the ABCD experimental code and behavioral data. The issues are as follows: different go stimulus durations across trials, the go stimulus is sometimes not presented, faulty SSD, different stop-signal durations for different SSDs, non-uniform conditional trial probabilities, trial accuracy incorrectly coded, SSD values start too short, and low stop trial probability. We judge these issues to vary from fundamental (e.g., different go stimulus durations across trials) to more minor (e.g., low stop trial probability). Indeed, some of the minor issues may reflect intentional design choices (e.g., low stop trial probability), but if so, we believe that those choices are suboptimal (for reasons that we lay out below). Additionally, we believe that the most fundamental issues are incontrovertible errors, and in the aggregate, we believe these eight issues significantly compromise the utility of the stopping data in this study for understanding the construct of response inhibition.

We structure this paper as a series of issues. We order the issues roughly from what we judge to be most to least fundamental. For each, we outline the issue, demonstrate its effects in the ABCD data, suggest prospective solutions to the study organizers for future data collections, and suggest retrospective solutions to data users who want to make the most of the existing data. Also, we propose three preliminary computational frameworks that may be able to capture the violations that result from the first design issue.

Since the original submission of this manuscript, a rebuttal preprint manuscript has been submitted from a group of authors involved in the design of the ABCD stop-signal task (*Garavan et al., 2020*). In short, we stand behind the conclusions of the present manuscript and found no evidence or argumentation within their rebuttal that meaningfully challenges any of our manuscript's conclusions; importantly, their manuscript did not dispute the presence of any of the specific issues raised below. We would like to thank *Garavan et al., 2020* for pointing out an error in our code, which we corrected on July 17, 2020 (https://www.biorxiv.org/content/10.1101/2020.05.08.084707v4).

Before we lay out the first issue, we will first break down the trial structure of the ABCD stop-signal task (see *Figure 1*). 5/6 of all trials are go trials, in which a subject sees a go stimulus (a rightward- or leftward-pointing arrow) and makes one of two speeded responses based upon the direction of the arrow. The go stimulus is removed from the screen after 1000 ms or when a response occurs, whichever comes first. On 1/6 of all trials, this go stimulus is replaced with the stop signal (a vertical arrow) after the go stimulus has been on the screen for the duration of the SSD or when a response occurs; the stop signal is then presented for 300 ms (but see Issue 4). Therefore, on go trials, the go stimulus is on the screen for 1000 ms or the response time (RT), whichever comes first, whereas on stop trials, the go stimulus is on the screen for SSD or RT, whichever comes first, and is never concurrently on the screen with the stop stimulus.

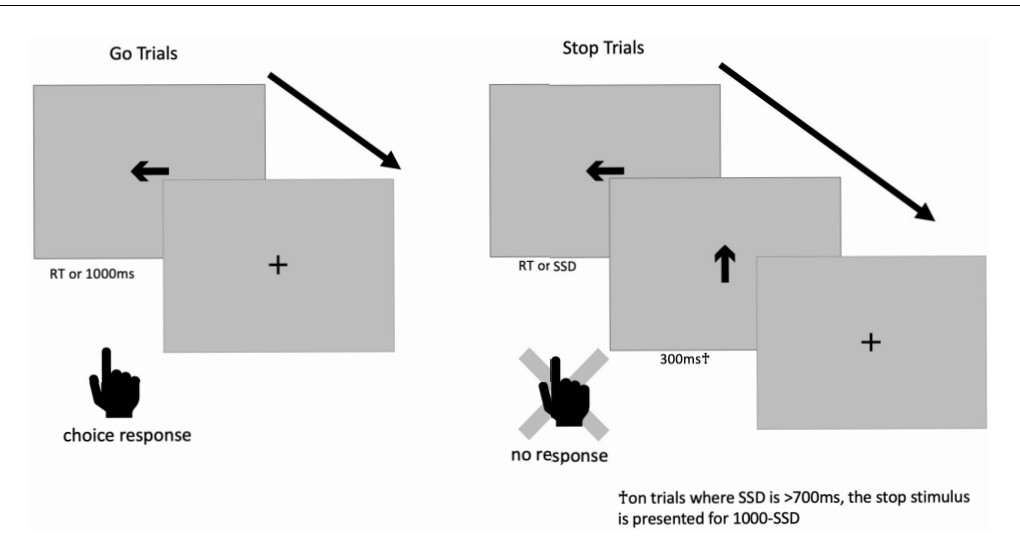

**Figure 1.** Stop-signal task trial structure.

## Results

### Issue 1: different go stimulus durations across trials

This brings us to our first and most fundamental issue: the go stimulus is presented for a much longer duration on go trials than on stop trials. Mean go stimulus duration on go trials was 569 ms (standard deviation [SD] = 105 ms), and mean go stimulus duration on stop trials was 228 ms

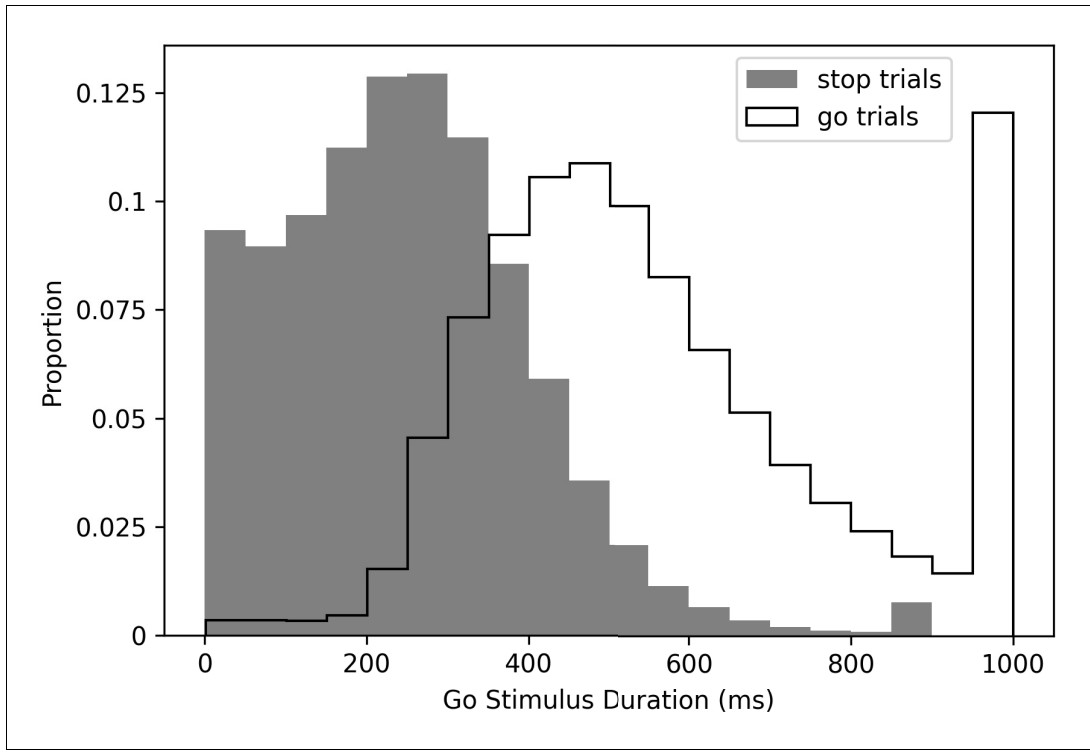

**Figure 2.** Proportion of go stimulus durations on go and stop trials.

(SD = 118 ms), so on average, subjects had 341 ms longer on go trials to apprehend the go stimulus (see *Figure 2* for full distributions).

This is important because the current models for understanding stopping performance require that the go process is the same on go and stop trials. The main dependent variable in the stop-signal task is stop-signal reaction time (SSRT), which quantifies the latency of inhibition, but the estimation of SSRT requires application of a model because there is no overt response associated with a successful stop and thus stopping latency cannot be directly measured. The independent race model assumes context independence, which means that the go process and its finishing time are not affected by the presentation of the stop signal. It also assumes stochastic independence, which means that the finishing times of the go and the stop processes are independent on any given trial. In this manuscript, we focus on the assumption of context independence. Context independence is essential for calculating SSRT because context independence allows one to assume that the full distribution of responses on go trials can stand in for the (unobservable) full distribution of go processes on stop trials. Additionally, violations of context independence contaminate other major dependent variables in the stop-signal task, including the inhibition function and stop-failure RT (*Bissett et al., 2021*; *Logan and Cowan, 1984*). Paradoxically, some existing models have suggested that an array of interacting excitatory and inhibitory neurons give rise to behavior that appears independent, perhaps because the interaction is brief but very strong (*Boucher et al., 2007*). However, here we exclusively focus on behavioral evidence of independence or lack thereof.

In both simple (e.g., *Froeberg, 1907*) and choice (e.g., *Kaswan and Young, 1965*) reaction times, there is evidence that shorter duration stimuli yield slower responses. This effect of stimulus duration on response speed occurs even for stimuli presented for hundreds of milliseconds (*Kahneman and Norman, 1964*; *Kaswan and Young, 1965*). This relates to Bloch's law (*Bloch, 1885*), which states that intensity and duration can be traded off for shorter duration stimuli, though this trade-off only occurs over the first 100 ms. For example, reducing stimulus duration by half is equivalent to reducing intensity by half. This also relates to Pieron's law, which states that RTs decrease with stimulus intensity (*Pieron, 1914*). Therefore, having shorter duration go stimuli on stop trials than go trials is akin to having a lower intensity go stimulus on stop than go trials, which slows RT. Taken together, a long history of work suggests that (all else being equal) shorter duration visual stimuli tend to yield slower responses.

In the ABCD stopping experiment, go stimuli are presented for much shorter durations on stop trials than on go trials, so the work described in the preceding paragraph suggests that the go process will be faster on go trials than on stop trials. However, in order to extract SSRT, one must make the assumption that the go process is the same on go and stop trials (i.e., context independence). Therefore, we expect that context independence is violated in the ABCD dataset, which would contaminate major dependent variables in the stop-signal task including SSRT estimates. Additionally, because go stimuli on stop trials are presented for a duration equal to SSD, the degree to which violations of context independence occur are likely to differ across SSDs. When SSD is short (e.g., 50 ms), context independence should be more severely violated because the difference in go stimulus duration between stop (e.g., 50 ms) and go (up to 1000 ms) trials is so large.

## Evidence for Issue 1 in ABCD data

The primary way to evaluate context independence is to compare reaction times on go trials to reaction times on stop-failure trials (but note that the prediction of the race model concerning faster stop-failure than go responses is conditioned on both context independence and stochastic independence, *Colonius and Diederich, 2018*). If the former is longer than the latter, then context independence is taken to hold (*Bissett et al., 2021*; *Logan and Cowan, 1984*; *Verbruggen et al., 2019*). On average across all subjects, stop-failure RT (M = 456 ms, SD = 109 ms) was shorter than overt responses on go trials (M = 543 ms, SD = 95 ms, 95% confidence interval of the difference [85.9, 88.8]). However, for 524 of the 8,464 subjects (6.2% of all subjects), mean stop-failure RT was longer than mean RT on overt responses in go trials, suggesting that a subset of subjects violated context independence. Though note that the comparison of stop-failure and go RT is a conservative measurement that will only show violations of context independence if they are severe (*Bissett et al., 2021*). Additionally, it will only reveal slowing of the go process on stop trials, even

though context independence could be violated by the go process being faster on stop than go trials (e.g., see our guessing model below).

In order to further evaluate whether the go process is impaired on stop trials as a result of the shorter go stimulus, we compared choice accuracy on stop-failure trials with choice accuracy on all overt (non-omission) go trials. Stop-failure trials had much lower accuracy (79%) than overt go trials (90%), 95% confidence interval of the difference (10.6%, 11.1%). Additionally, choice accuracy on stop-failure trials was increasingly impaired at shorter SSD (see *Figure 3*). In other datasets, choice accuracy on stop-failure trials, even at shorter SSDs, tends to be similar to overt go trials (*Bissett et al., 2021*), which suggests that the impaired go accuracy on stop-failure trials in the present study results from the shorter go stimulus durations on stop trials in this task. Therefore, this lower choice accuracy is consistent with the go process being fundamentally impaired on stop trials compared to go trials, particularly at short SSDs, violating the assumption of context independence. Additionally, these violations of context independence manifest in an unprecedented way (i.e., lower choice accuracy on stop failure trials), showing that Design Issue 1 drives a new form of violations that go beyond any previous evidence for violations in stop-signal tasks (e.g., *Bissett et al., 2021*).

## Potential mechanisms underlying Issue 1

New computational models need to be developed to capture the Design Issue 1 violations in the ABCD data. Issue 1 embeds violations in the essential fabric of their task design, and existing models do not provide guidance on how to excise these violations or limit their contamination of the ABCD data. To our knowledge, all existing models for stopping assume that the parameters that generate the distribution of go processes take the same value on go and stop trials (e.g., *Boucher et al., 2007*; *Logan et al., 2014*; *Logan et al., 2015*; *Matzke et al., 2017*). Therefore, there is no existing model that can capture the ABCD stopping data and no valid way to compute SSRT from these data because the go process is different on go and stop trials (see *Figure 3*). This is most clear on 0 ms SSD trials, in which the go stimulus is not presented at all (see Issue 2), so subjects must be guessing and therefore the go process must be fundamentally different than when it is stimulus driven. Additionally, existing simulations that suggest SSRT estimates are robust to violations of independence (*Band et al., 2003*) do not evaluate the type of contamination introduced by Issue 1, namely a different set of processes generating the go process on go and stop trials. Instead, they show that SSRT is relatively robust to correlations between go RT and SSRT or correlations between SSD and SSRT, which are unrelated to Design Issue 1.

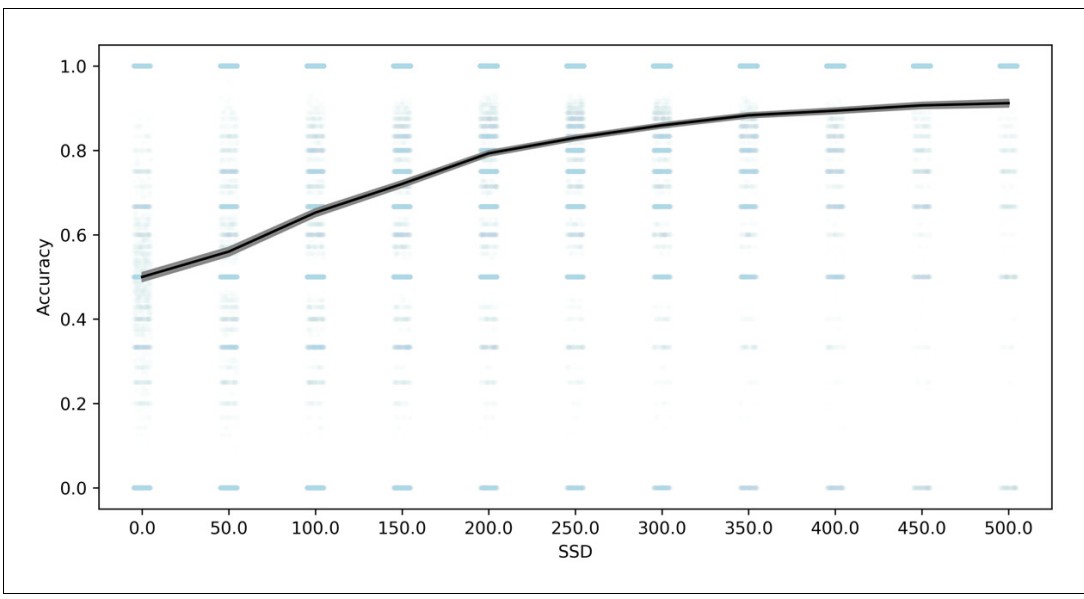

**Figure 3.** Choice response accuracy on stop-failure trials across SSD. Note: 95% confidence intervals are presented as gray confidence bands. Individual-subject datapoints are presented in blue with lower alpha.

There is evidence that violations of context independence can occur even without the introduction of Design Issue 1 (*Akerfelt et al., 2006*; *Bissett et al., 2021*; *Colonius et al., 2001*; *Gulberti et al., 2014*; *Ozyurt et al., 2003*); however, we believe that the violations in the ABCD dataset go beyond this existing work and will require distinct computational solutions for the following reasons. First, these previous violations occur in spite of carefully equated exogenous stimulus parameters, whereas the ABCD design encourages violations by presenting different exogenous stimulus parameters between go and stop trials. Second, as we mentioned above, the violations in the ABCD dataset manifest in an unprecedented way (i.e., lower choice accuracy on stop-failure trials) that go beyond previous evidence for violations (e.g., *Bissett et al., 2021*). This suggests that the generating mechanisms underlying violations in ABCD are fundamentally different than in other stopping datasets. Theoretical explanations that aim to account for violations, such as failures to trigger the stop process (*Matzke et al., 2017*) or variability in the speed or potency of inhibition (*Bissett et al., 2021*), focus on modifications to the stop process across trials. In contrast, greatly reduced choice accuracy on stop-failure trials compared to go trials indicates that the go process is modified across go and stop trials in the ABCD dataset. Therefore, even if future computational modeling efforts account for violations in datasets without design Issue 1, we do not believe that they will generalize to the violations in the ABCD, given that the violations in ABCD are driven by the idiosyncrasies of Issue 1.

In order to begin the process of developing a new theoretical framework for understanding the violations of context independence in the ABCD dataset, we propose three computational models that aim to capture the violations in the ABCD dataset. All three instantiate adjustments to the independent race model (*Logan and Cowan, 1984*). First, the *slowed go processing* model suggests that the drift rate for the go process, which measures the speed of information processing, is slowed at shorter SSDs. We justify this because weaker exogenous drive from a shorter stimulus should yield slower and less effective information processing of that stimulus. Second, the *guessing* model suggests that the go process on stop trials is a mixture of guesses and stimulus-driven responses, such that all guesses at 0 ms SSD stop trials gradually give way to all stimulus-driven responses (i.e., no guesses) as SSDs lengthen. We justify this because guesses are necessary at 0 ms and choice accuracy of stop-failure trials at shorter SSDs remains very low as if subjects are mostly randomly guessing. Third, the *confusion* model suggests that subjects may be confused on how to respond at short delays, impairing both the go and the stop processes, which we instantiate as slower drift rates for both the go and the stop processes at shorter SSDs. We justify this because short SSD trials may violate subjects' basic expectation of a clear and perceptible go stimulus that precedes the stop signal on stop trials, resulting in confusion and weaker drive to both behavioral responses. All three of these models can naturally explain the higher choice error rate on stop-failure trials at short SSDs than long SSDs, as reducing drift rate reduces accuracy and guesses have chance accuracy. All three also roughly match the empirical inhibitory function from the ABCD data (see *Figure 4*), though only the third confusion model captures the non-monotonicity at the 0 ms SSD. For these reasons, they are promising, though preliminary, mechanistic models to explain the violations of context dependence in the ABCD data. Fully establishing these models will require substantial additional work, including parameter fitting and recovery, model recovery, and model comparison.

With these caveats in mind, we simulated data to estimate the degree of SSRT contamination that would arise from applying the independent race model (*Logan et al., 2014*; *Logan and Cowan, 1984*) and assuming context independence if instead each of these three models are the correct generating model for the go process on stop trials. We fit the observed grand mean go RT and SSRT distributions to get plausible estimates of each value, then we adjusted the pertinent parameters (e.g., slow the drift rate of the go process at short SSDs in the slowed go processing model) according to the three models specified above (see Materials and methods for additional details). Finally, we computed an SSRT for simulated data generated from the independent race model and each of the three proposed models above while making the assumption of context independence (which we know to be violated in our three alternative models). This would overestimate mean SSRT by 75 ms for the slower drift rate model (slower drift rate SSRT = 207 ms and independent race SSRT = 282 ms), underestimate SSRT by 61 ms with the mixture guessing model (mixture guessing = 343 ms), and overestimate mean SSRT by 22 ms for the confusion model (confusion SSRT = 260 ms). Additionally, the degree of estimation errors varies with SSD (see *Figure 5*). Therefore, mean SSRT estimates would be contaminated if any of these three models capture the go

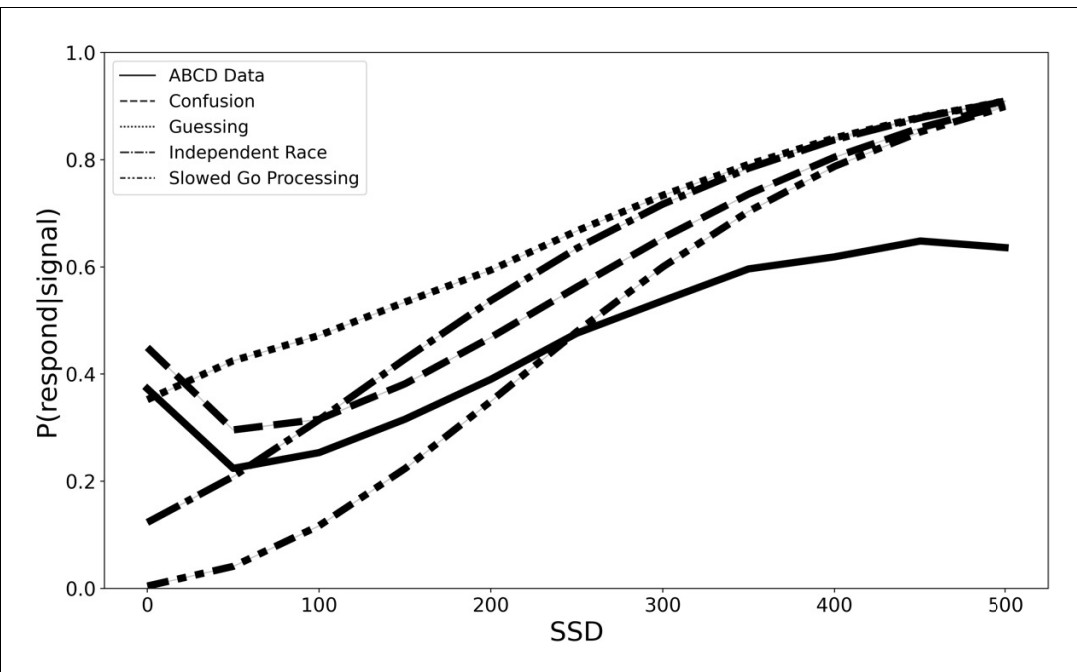

**Figure 4.** Inhibition functions from the real ABCD data and four simulated models.

process on stop trials, but the researcher instead assumed context independence, as the SSRT estimation procedure requires. These models instantiate differences in the go process between stop and go trials (i.e., context dependence), but by assuming context independence, these differences in the go process are incorrectly ascribed to the stop process, contaminating SSRT estimates.

However, as *Garavan et al., 2020* point out, 'Examining individual differences is a central goal of the ABCD study. Crucially, for the utility of the ABCD SSRT measure to be degraded as a measure

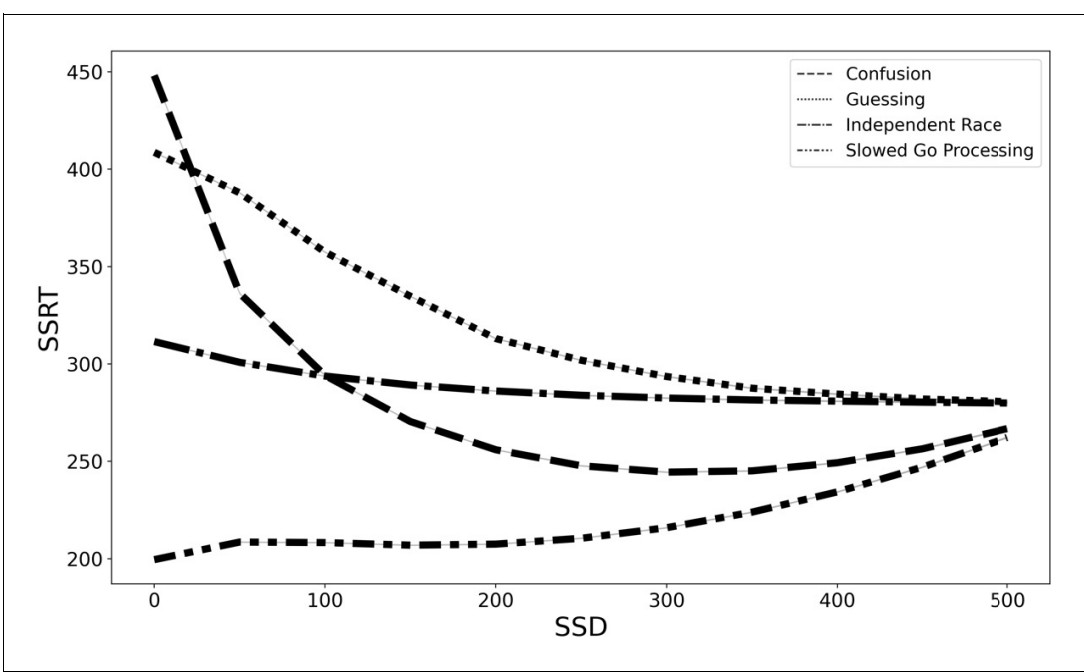

**Figure 5.** SSRT estimates from four generating models (slowed go processing, guessing, confusion, and independent race) across SSDs if we assume context independence.

of individual differences, violations of context independence must result in more than a shift in mean SSRT. Rather, the rank order of participants' SSRT values must be substantially altered'.

To examine individual differences, we created 8,207 simulated subjects that shared features from the 8,207 real ABCD subjects. We simulated go RTs based on real ABCD subjects' performance and implemented three ways to determine SSDs in the simulation (ABCD weighted, fixed SSDs, and simulated tracking SSDs). Given the observed violations of context independence, we do not have trustworthy estimates of individual-subject SSRTs. In order to assign an SSRT value to each simulated subject, we sampled randomly from an SSRT distribution with a mean that equaled the observed ABCD grand mean but assumed four different amounts of between-subject variability (ranging from SD = 0–85 ms). This range of between-subject variability was informed by evaluating the 20 simple stopping conditions from a recent large-scale stopping study which had a mean between-subject SD of SSRT = 43 ms with a range of 28 ms–85 ms (*Bissett et al., 2021*). For each simulated subject, we computed SSRT (assuming context independence) separately based upon data generated from our four generating models (slowed go processing, guessing, confusion, and independent race). We then computed rank correlations of SSRT estimates for these 8,207 simulated subjects between the independent race model and each of our three proposed models, for our three SSD determination methods (ABCD weighted, fixed, and tracking SSD) and four values of between-subject variance in assumed SSRT (85 ms, 25 ms, 5 ms, and 0 ms) (see *Table 1* for Results and Materials and methods for additional simulation details).

Rank correlations quantify the degree that SSRT is degraded as a measure of individual differences if we assume the independent race model when a different generating model (like our three proposed models) more appropriately characterizes the underlying data generating mechanism in the ABCD stopping task. If we assume high between-subject variability in SSRT (SD = 85 ms), then the rank correlations are largely preserved across generating models and SSD approaches (r range = 0.85–0.99, mean = 0.93). However, as between-subject variability in SSRT reduces (SD = 25 ms), the rank correlations decrease (r range = 0.66–0.97, mean = 0.78). At very low between-subject variability, (SD = 5 ms) the rank correlations can become very low (r range = 0.18–0.89, mean = 0.63). At the limit, if we assume no between-subject variability in SSRT and therefore between-subject variance is driven entirely by differences in SSD, go RT, and the different generating models, rank correlations continue to reduce but remain well above 0 on average (r range = 0.02–0.91, mean = 0.54). This suggests that differences in SSD distribution or go RT across subjects can inflate estimates of SSRT rank correlations in our simulations, and therefore SSRT individual differences may be more contaminated than the above rank correlations suggest. Taken together, these results show that design Issue 1 can degrade individual differences, with the degree of misestimation increasing as between-subject variability in SSRT decreases.

To reiterate, these are only preliminary model proposals and require more rigorous scrutiny. However, no existing models for stopping can capture the context dependence that is embedded in the fabric of the ABCD data by Design Issue 1. These preliminary simulations suggest that mean SSRT estimates are contaminated and individual differences may be contaminated by the Design Issue 1 violations of context independence, particularly if the true between-subject variability in SSRT is small.

**Table 1.** Rank correlations of SSRTs from the three alternative generating models (rows) with the independent race model across the three SSD determination methods (ABCDw = ABCD weighted, fixed, and tracking) and four SSRT standard deviation (SD) scales (85 ms, 25 ms, 5 ms, and 0ms).

| | SD = 85ms (mean:0.93) | | | SD = 25ms (mean:0.78) | | | SD = 5ms (mean:0.63) | | | SD = 0ms (mean:0.54) | | |
|---|---|---|---|---|---|---|---|---|---|---|---|---|
| | ABCDw | Fixed | Tracking | ABCDw | Fixed | Tracking | ABCDw | Fixed | Tracking | ABCDw | Fixed | Tracking |
| **Confusion** | 0.848 | 0.973 | 0.995 | 0.73 | 0.89 | 0.965 | 0.82 | 0.893 | 0.554 | 0.847 | 0.903 | 0.021 |
| **Slowed Go Processing** | 0.932 | 0.976 | 0.949 | 0.661 | 0.896 | 0.677 | 0.178 | 0.887 | 0.27 | 0.093 | 0.895 | 0.183 |
| **Guessing** | 0.883 | 0.907 | 0.951 | 0.713 | 0.765 | 0.754 | 0.813 | 0.88 | 0.342 | 0.848 | 0.913 | 0.176 |

## Prospective suggestions for Issue 1

In order to address Issue 1, we would recommend that the ABCD study organizers present the go stimulus for a fixed period of time on every trial, perhaps 1000 ms. When a stop signal occurs, it should not replace but it should be presented in addition to the go stimulus. If the study designers would like to keep all stimuli in the center of the screen, they could superimpose the stop stimulus around the arrow (e.g., a circle). However, as suggested by a reviewer, to avoid perceptual interactions the stop circle should be >1 degree of visual angle from the go arrow. Therefore, the go stimulus would be identical in form, size, and duration for all go and stop trials. This should eliminate any possibility that different go durations drive different go processes, violating context independence and contaminating dependent variables.

## Retrospective suggestions for Issue 1

The main reason that we have suggested that Issue 1 is the most fundamental is if one assumes that shorter go stimuli on stop trials yield slower, impaired go processes when compared with the longer go stimuli on go trials, which we believe is reasonable assumption given over 100 years of RT research (*Bloch, 1885*; *Froeberg, 1907*; *Kahneman and Norman, 1964*; *Kaswan and Young, 1965*; *Pieron, 1914*), then context independence is violated and major dependent variables in the stop-signal task are contaminated. Additionally, the violations of context independence, as measured by a decrease in choice accuracy on stop-failure trials, appear to go beyond existing evidence of violations in stopping data (*Bissett et al., 2021*). These violations may also influence common neuroimaging task contrasts like stop success versus go and stop failure versus go, as this introduces additional differences between trial types, including go stimulus duration and go stimulus reaction time, that will contaminate the ability of the contrast to isolate processes of interest like response inhibition. Given the above empirical evidence and simulations, and as suggested by our reviewers, unless the ABCD community shows that this design issue does not distort conclusions based upon SSRT estimates (or any other stop-signal measure), researchers should not use the ABCD dataset to estimate SSRTs and should use the neuroimaging data with caution.

We also suggest two practical suggestions to attempt to avoid some of the most extreme violations, but we note significant shortcomings of these solutions. First, in line with consensus guidelines (*Verbruggen et al., 2019*), we would recommend removing the subjects who have severe violations as evidenced by mean stop-failure RT > mean no-stop-signal RT (524 of the 8,464 that we analyzed, 6%). However, this may impact the representativeness of the sample. Also, though this may eliminate some of the subjects who most severely violated, violations are not an all-or-none phenomenon and this will likely leave in subjects who violate, albeit to a less extreme degree. Additionally, our above simulations demonstrate that violations of context independence can manifest as go slowing (as in slowed go drift rate) or go speeding (as in fast guesses at short SSDs), the latter of which would not be captured by this criterion at all. Therefore, this first suggestion should not be taken as sufficient to confidently eliminate the influence of violations. Second, given our suggestion that violations may be most severe at short SSDs (because the difference in go stimulus duration between go and stop trials is maximal at short SSDs, see *Figure 3*), and given work suggesting violations of context independence may be most severe at short SSDs (*Akerfelt et al., 2006*; *Bissett et al., 2021*; *Colonius et al., 2001*; *Gulberti et al., 2014*; *Ozyurt et al., 2003*), we would recommend that any results be verified when only longer SSDs are used, perhaps only SSDs $\geq$ 200 ms. However, note that removing stop trials that have SSDs < 200 ms would remove 51.5% of all stop trials in the ABCD dataset, leaving less than an average of 30 stop trials per subject. Additionally, *Figure 3* shows that choice accuracy on stop-failure trials does not asymptote at 200 ms, so a longer cutoff (and additional data loss) would more confidently avoid severe violations. Neither of these two suggestions resolve Issue 1, but they may eliminate the subjects and trials most likely to show severe violations, respectively.

In order to resolve Issue 1, the different go durations that encourage context dependence must be eliminated (see prospective suggestions), or new models for stopping must be developed to accommodate context dependence, the latter of which we consider to be of utmost importance to advancing the stop-signal literature. We attempt to make preliminary progress in this direction with the models that we instantiate in the above simulations and in companion work that suggests preliminary theoretical frameworks that can account for context dependence (*Bissett et al., 2021*).

However, these are very preliminary proofs of concept and much more work is necessary to fully flesh out a viable model for stopping that can accommodate context dependence. Additionally, we believe that Design Issue 1 introduced idiosyncratic violations (i.e., reduced choice error rates) that will require idiosyncratic computational solutions.

## Issue 2: go stimulus sometimes not presented

Issue 2 can be seen as a special case of Issue 1. When the stop signal occurs, it replaces the go stimulus, and SSD can reduce to 0 ms. When the SSD is 0 ms, a go stimulus does not occur on the stop trial. This is an issue because extant models of stopping assume a race between two processes (*Logan and Cowan, 1984*), and there is not a go stimulus to initiate a go process or to drive one of the two choice responses. The absence of a go stimulus may make successfully stopping trivial (as the go process never started) or may confuse subjects given these trials have a fundamentally different structure (i.e., one stimulus instead of two).

### Evidence for Issue 2 in ABCD data

Nine percent of all stop trials had an SSD of 0 ms (see *Figure 6*). In a typical stop-signal experiment, a 0 ms SSD trial means that the go and stop stimuli onset at the same time and are presented concurrently for a period of time. This tends to produce a very high stop success rate at a 0 ms SSD. In these data, we found that the stop success rate at the 0 ms SSD was 60.1%. This surprisingly low value may have resulted from subjects being confused as to how to approach these trials (see the confusion model above). It also may have been driven by subjects who stopped inhibiting their responses. Additionally, see Issue 3 for a contribution explanation.

### Prospective suggestions for Issue 2

The suggestion from Issue 1 will also naturally resolve Issue 2. When SSD is 0 ms, the arrow go stimulus would be presented at the same time as a secondary stop signal presented around it.

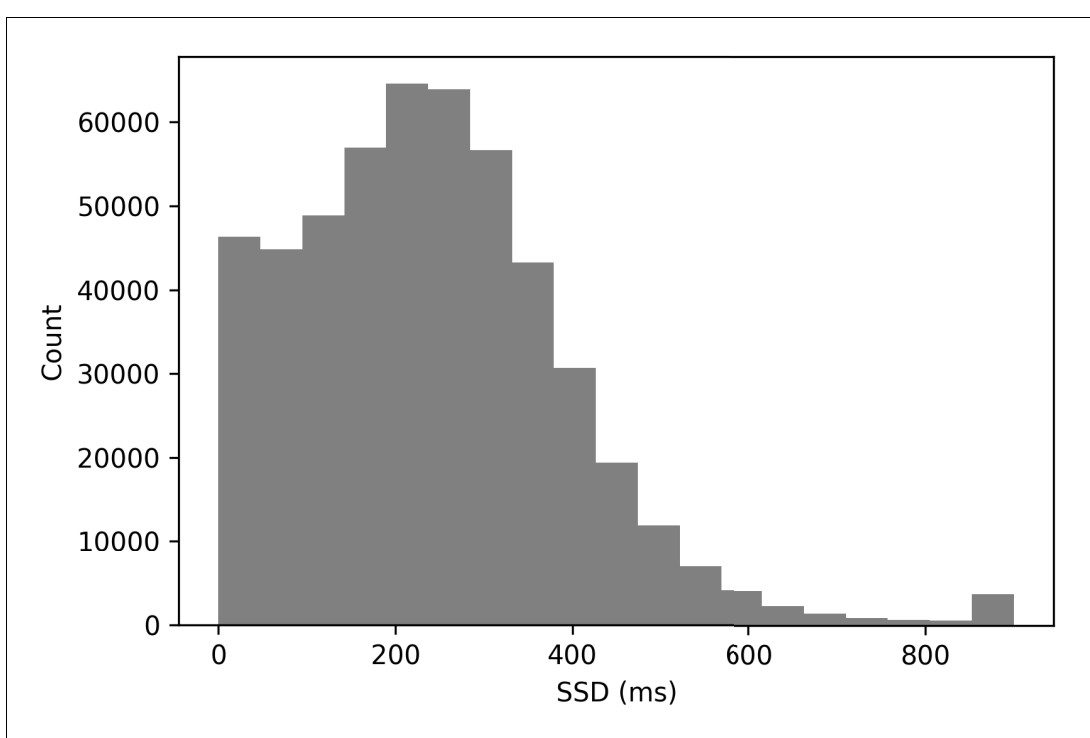

**Figure 6.** Histogram of stop-signal delays.

### Retrospective suggestions for Issue 2

Given there is not a go stimulus to drive a choice toward a specific go response on 0 ms SSD trials, we recommend that these 0 ms SSD trials be removed from any behavioral or neuroimaging analyses. The complete lack of a go stimulus is at least as fundamental as 1, but this only occurs on 9% of all stop trials, so this issue can be addressed by removing these trials.

## Issue 3: faulty SSD

The ABCD study uses the common 1-up/1-down tracking algorithm (*Levitt, 1971*) to determine SSD, which involves increasing SSD by 50 ms whenever a subject successfully stops and decreasing SSD by 50 ms whenever a subject fails to stop. However, if SSD is 50 ms and a subject makes a response that is faster than 50 ms (e.g., 25 ms), then this triggers a glitch in the experiment code in which all subsequent stop trials have a response erroneously recorded at that same timestamp (e.g., 25 ms). Therefore, all subsequent stop trials are treated as stop failures (because this one initial response is recorded for all subsequent stop trials), and SSD remains stuck at the minimum of 0 ms for the remainder of that subject's stopping dataset.

### Evidence for Issue 3 in ABCD data

The triggering condition for this coding error is very specific and occurs somewhat rarely. In the entire sample, only 2.7% of subjects had this specific problem. However, this issue interacts with Issue 2 and partially explains the low stop accuracy at 0 ms SSD. Once these trials are excluded, the stop accuracy at 0 ms increases from 60.5% to 62%.

### Prospective suggestions for Issue 3

This appears to be a coding error, so we would recommend that the error be resolved by ensuring that any response on a given trial does not propagate forward incorrectly as also being a response with the same duration in subsequent trials.

### Retrospective suggestions for Issue 3

Given the rarity of this problem, in addition to the severity of its manifestation (i.e., it ensures that every subsequent stop trial is recorded as a stop-failure with a 0 ms SSD, irrespective of actual behavior), we recommend removing the 2.7% of subjects who trigger this issue.

## Issue 4: different stop signal duration for different SSDs

The independent race model (*Logan et al., 2014*; *Logan and Cowan, 1984*) not only assumes *go* context independence, but also *stop* context independence, which means that the stop process is the same across SSDs. However, in an issue that mirrors Issue 1, the stop signal is presented for different durations at different SSDs. If SSD is $\leq$ 700 ms, then the stop signal is presented for 300 ms. If SSD is >700 ms, then the stop signal is presented for 1000-SSD, with a maximum SSD of 900 (and therefore a minimum stop duration of 100 ms).

### Evidence for Issue 4 in ABCD data

SSDs > 700 ms are rare in these data (see *Figure 6*, 1.1%). This rarity makes it difficult to compute SSRT exclusively at these longer delays to evaluate whether the stop process may be changed as a result of the stop signal being presented for a shorter duration at the longest SSDs.

### Prospective suggestions for Issue 4

We recommend a fixed stop-signal duration across all SSDs, perhaps 300 ms.

### Retrospective suggestions for Issue 4

Given that SSDs that are >700 ms are so rare (1.1%), we suggest removing them from any analyses. This will eliminate their contamination on any averages without resulting in significant data loss.

## Issue 5: non-uniform conditional trial probabilities

In stop-signal tasks, the default way to determine trial sequences is to randomly or quasi-randomly (e.g., random without replacement from a fixed pool of trials to ensure the same number of stop trials in each block) select which subset of trials will include a stop signal. However, in fMRI task-based experiments, it is common to select trial sequences in order to optimize power of finding an effect or a difference between contrasting conditions (*Durnez et al., 2017*; *Kao et al., 2009*; *Liu et al., 2001*). Increasing detection power can be achieved by adjusting the conditional probabilities such that trials are more likely to follow the same trial type than change to a new trial type (i.e., the design becomes more block-like). However, this push for greater power needs to be weighed against known expectancy effects in the stop-signal task (*Bissett and Logan, 2012b*), which can change the processes involved in a task when subjects can predict what is coming. Indeed, some modern software (*Durnez et al., 2017*) allows the researcher to explicitly trade off the competing goals of power and avoiding trial-by-trial contingencies.

The ABCD stopping study has a highly non-uniform distribution of transition probabilities between go and stop trials. Given the overall probability of a stop signal is 1/6 or 0.167 and a uniform conditional probability across trials, the probability of a stop signal given the immediately preceding trial was a stop signal should be 0.167. In the ABCD task, this value is 0, as stop trials never repeat. Additionally, stop trials almost never occur if there was a stop trial two trials before (conditional probability of 1.8%). We plot the conditional probability of a stop signal given the most recent stop signal, both in the ABCD data and for an example where the probability of a stop signal was equal across all trials (see *Figure 7*). Subjects have the potential to learn this contingency that stop trials never repeat and seldom occur after one intervening go trial. This may be important because usually go reaction time slows after stop signals (post-stop-signal slowing, *Bissett and Logan, 2011*; *Bissett and Logan, 2012a*; *Rieger and Gauggel, 1999*), but if subjects learn that stop signals are never followed immediately by another stop signal then this results in go reaction time speeding (*Bissett and Logan, 2012b*). Therefore, by making such drastic adjustments to the conditional probabilities, this may change the way that subjects balance going and stopping across trials.

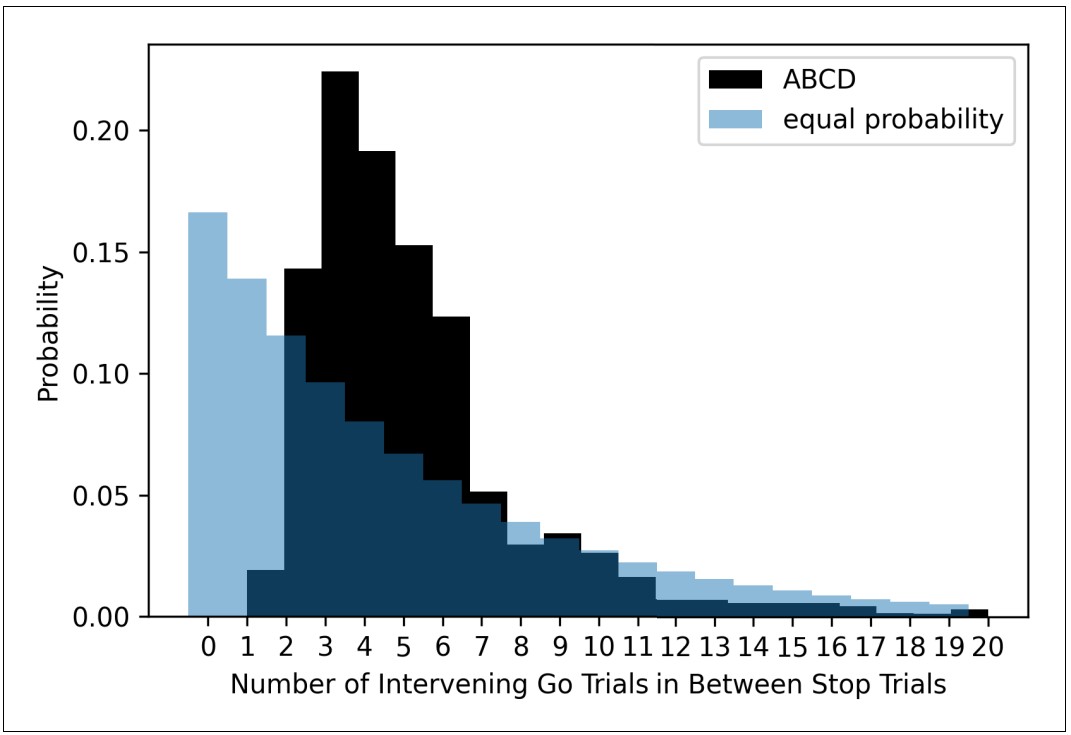

**Figure 7.** The probability of different numbers of intervening go trials between successive stop trials in the ABCD dataset (dark shading) versus expected probability distribution if stop trials were equally probable on all trials (light blue shading).

## Evidence for Issue 5 in ABCD data

In order to investigate whether subjects are learning this trial contingency and using it to change behavior, we broke the stop trials in the task into quantiles over time (i.e., first 15 stop trials, second 15, third 15, and fourth 15) and computed post-stop-signal RT changes in each. If subjects are learning this contingency and changing their behavior in response to it, we should see post-stop-signal slowing (slower RT on the trial immediately following a stop signal compared to RT on the trial immediately preceding a stop signal) reduce over the experiment, perhaps giving way to post-stop-signal speeding (faster RT on the trial immediately following a stop signal compared to RT on the trial immediately preceding a stop signal) toward the end of the experiment. This is not what we found (see *Figure 8*) as post-stop-signal slowing was around 20 ms for each quartile except the second, in which it was 40 ms. Therefore, we did not see our expected reduction of post-stop-signal slowing over time or post-stop-signal slowing in the early quartiles giving way to post-stop-signal speeding in the later quartiles. It appears that in this baseline session, subjects did not learn this contingency or did not change their behavior in response to it.

## Prospective suggestions for Issue 5

In order to eliminate the possibility of subjects learning this contingency and changing their behavior, we suggest that the ABCD study make the conditional probabilities fully uniform or at least much more uniform, most importantly allowing for the possibility of immediate repeats. Though learning effects were not apparent in the baseline session, subjects may learn this contingency and implement more severe behavioral changes in subsequent sessions as they have greater exposure to this contingency. Additionally, as subjects mature, they may develop the ability to learn this contingency and use it to change their behavior.

## Retrospective suggestions for Issue 5

The two trials after a stop signal never or virtually never include a stop signal, so they could have involved fundamentally different processes from normal go trials. In essence, if this contingency is

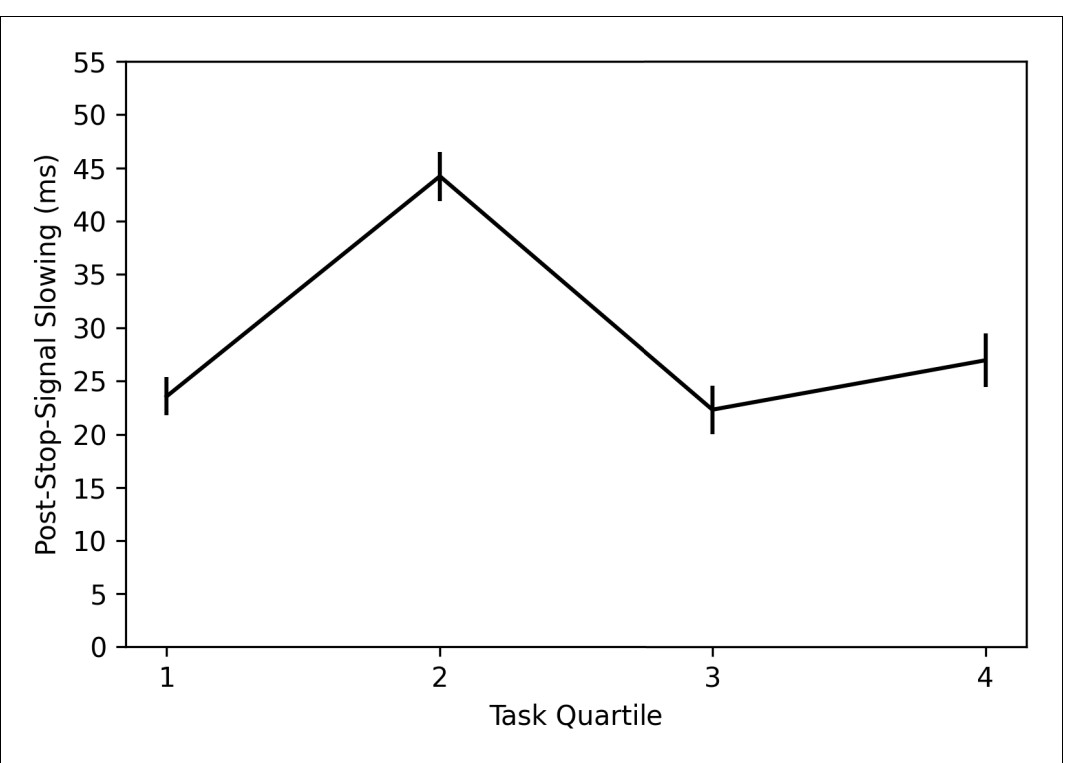

**Figure 8.** Post-stop-signal slowing across the four quantiles of the ABCD stopping data. Note: Error bars are 95% confidence intervals.

learned, the trial after a stop trial becomes a 'certain-go' trial, which has been shown to have different behavioral performance and neural responses (*Chikazoe et al., 2009*). However, we did not find evidence that subjects learned and adjusted their behavior based upon this contingency in this baseline session. Additionally, removing the two go trials after a stop signal would involve huge data loss (1/3 of all trials), so this does not seem warranted. Therefore, we do not think that Issue 5 will affect data users, at least in the baseline session.

## Issue 6: trial accuracy incorrectly coded

In the stop-signal task, there are two types of trials, go and stop, and each can be correct or an error. On a go trial, subjects can be correct by making the appropriate choice response or make an error by either making the incorrect choice response or by omitting their response. On stop trials, subjects can be correct by omitting their response or be incorrect by making a choice response. In three of four of these trial outcomes, there are errors in how the ABCD data categorizes trials. Additionally, stop-failure trials can be correct or incorrect with respect to the go stimulus, and this should be coded properly.

### Evidence for Issue 6 in ABCD data

All 247,786 trials that are categorized as incorrect stops have one or more overt responses, so these all appear to be categorized correctly in the output. 25,854 of the 376,479 trials (6.9%) classified as incorrect go trials appear to actually have the correct go response. The majority of these (25,343, 6.7%) can be attributed to a known issue, where response buttons were flipped, resulting in responses being recorded as incorrect, when they were in fact correct. We were unable to ascertain why the remaining 511 trials (0.01%) trials were miscategorized. 61,215 of the 2,162,718 trials (2.8%) classified as correct go trials are incorrect go trials. 4,208 (0.002%) of these can be attributed to the response button flip mentioned above. The other 57,007 (2.6%) instances appear to result from overwriting the first response with any subsequent response that is recorded later in the trial, which we understand to be inappropriate for speeded tasks generally and perhaps especially for a stop-signal task in which one wants to measure the speed of the first go process that completes, not a subsequent go process. Finally, 1,081 of the 260,053 trials (0.4%) classified as correct stops are incorrect stops, as they include one or more overt responses. This final class of errors results in SSD increasing by 50 ms when it should instead decrease by 50 ms. 154 of these 1,081 appear to result from a response occurring at the exact millisecond that one of the constituent stimuli occur in the trial. We were unable to ascertain why the other 927 of the 1,081 were recorded as correct stops, given there was one or more recorded responses on these stop trials.

### Prospective suggestions for Issue 6

Issue 6 appears to result from errors in the experimental code. Therefore, Issue 6 should be fixable by adjusting the experimental code. For example, in their code, there are three constituent periods within a stop-signal trial (the SSD, the 100–300 ms in which the stop-signal is on the screen, and the fixation period after the stop signal), and if the recorded RT = 0 ms in all three periods (either because there was no response or if it occurred at the exact millisecond that the stimulus was presented), then trials are categorized as a correct stop. However, on 154 'correct stop' trials, there is a response at the exact millisecond that one of these stimuli is presented. It is still recorded as a response in the output, so instead of categorizing trials and increasing SSD by ensuring RT = 0 ms outputs in each period, the trials should be categorized as a correct stop if no response is recorded in any of these three periods.

### Retrospective suggestions for Issue 6

For data users, Issue 6 can largely be resolved by recategorizing each trial type (which we did for all analyses within this work), according to the rules specified in the paragraph where we introduced Issue 6, before analyzing the data. However, as we briefly mentioned above, one effect of this error cannot be undone because 1,081 'correct stop' trials have overt responses, so they should have been categorized as incorrect stop trials and therefore SSD should have reduced by 50 ms instead of being increased by 50 ms. However, given how infrequent these are (0.4% of all correct stops),

this is unlikely to have a significant effect on the SSD tracking algorithm, and therefore we judge this issue to be more minor. These 1,081 should be recategorized as incorrect stops before analysis.

## Issue 7: SSD values start too short

The 1-up/1-down SSD tracking algorithm (*Levitt, 1971*) is an efficient way to sample the intermediate part of the inhibition function, which is the most informative for constraining SSRT estimates (*Band et al., 2003*; *Verbruggen et al., 2019*). In order to sample the intermediate part of the inhibition function for the maximal number trials, the SSD should begin at a value such that go RT = SSD + SSRT. For example, if the expected mean RT in the sample is 500 ms and the expected SSRT is 250 ms, then the SSD should start at 250 ms. Otherwise, the initial stop trials will have a high stop success rate, which is less informative for constraining estimates of SSRT and could drive strategy shifts in the experiment (e.g., subjects might de-emphasize the stop process at the beginning of the experiment if they recognize that any stop signals will occur with such a short SSD that even a slower stop process could beat the go process).

### Evidence for Issue 7 in ABCD data

In the ABCD dataset, SSD starts at 50 ms. We plot the stop accuracy and SSD (see *Figure 9*) across each of the 60 stop trials for all subjects. This shows that stop accuracy tends to be high (and SSD low) in the first ~10 stop trials but then stabilizes around 0.5, as desired, for the remaining ~50 stop trials. The one salient exception is trial 31, which is the first trial of their second block or session (some subjects do both blocks back-to-back and others do one block in one scanning session and the second block in a subsequent scanning session), which has a very low stop accuracy.

### Prospective suggestions for Issue 7

In order to create a more efficient design that eliminates this initial part of each section when many subjects have high stop success rates that are less informative for SSRT estimates, we recommend starting each session with a more canonical 250 ms SSD. This is similar to the mean SSD (230 ms) and final SSD (265 ms) across subjects, so should sample the intermediate part of the inhibition function from the start for the most possible subjects.

### Retrospective suggestions for Issue 7

It is unclear whether subjects are recognizing these very short SSDs in the beginning of each session and adjusting their performance or strategies in response to them. For example, as suggested above, subjects could de-emphasize the stop process at the beginning of each session. Therefore, we would recommend a cautious approach of ensuring that any conclusions do not qualitatively change when initial trials (perhaps the first seven stop trials with stop success rates > 0.6) are removed from analyses. An even more cautious approach would be to remove trials up to trial 15, as this is the approximate point that the stop-signal accuracy asymptotes.

## Issue 8: low stop trial probability

Stop-signal probability is commonly 0.25, 0.33, or a similar value in stop-signal studies (*Verbruggen et al., 2019*). Increasing stop-signal probability increases power for SSRT calculation, stop-based imaging contrasts, and other analyses (e.g., the inhibition function *Logan and Cowan, 1984*). Values greater than ⅓ are somewhat rare, perhaps because there is a belief that high stop-signal probabilities may fundamentally change the stop process (but see *Bissett and Logan, 2011*; *Logan and Burkell, 1986*).

### Evidence for Issue 8 in ABCD data

This ABCD study uses a 0.167 stop probability, which is unusually low.

### Prospective suggestions for Issue 8

We suggest increasing the stop probability to 0.25, which would put this data acquisition more in line with the literature and would provide 50% more stop trials to increase analytical power without increasing the length of the data acquisition. We do not know of any research that suggests any benefits of any kind when reducing stop probability to as low as 0.167.

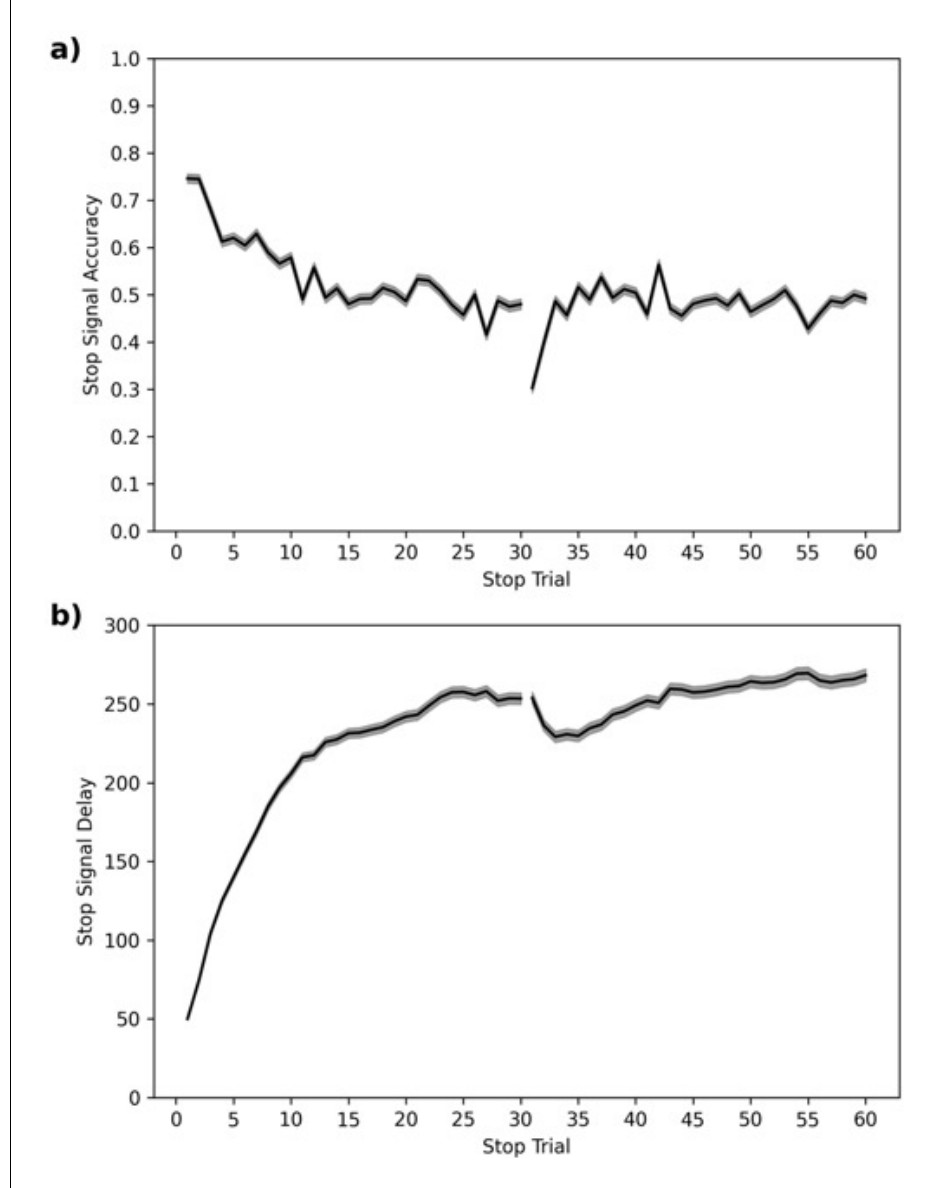

**Figure 9.** Average stop-signal accuracy (**a**) and stop-signal delay (SSD, **b**) across subjects for each of the 60 stop trials. Note: 95% confidence intervals are presented as gray confidence bands.

## Retrospective suggestions for Issue 8

We do not know of any published evidence that suggests that low stop trial probability may contaminate the data. Therefore, we do not believe that Issue 8 will affect data users, except that the low stop trial probability means that there are fewer stop signals to constrain and provide accurate estimates of stopping performance.

## Discussion

The ABCD study is an unprecedented investment toward understanding human cognition and brain development, and the stop-signal data is a key part of this dataset as one of only three tasks that are acquired within each neuroimaging session. Above, we laid out eight issues starting with more fundamental and moving towards more minor that, when taken together, we believe to significantly undermine existing ABCD stopping data. We have not evaluated the emotional N-back or monetary incentive delay tasks, but we hope that future work will review the experimental code and data from

these tasks. We hope that our manuscript shines a light on the importance of quality assurance of task design, particularly for large-scale studies. We would also like to note that at least some of these design issues extend to the IMAGEN study (*Schumann et al., 2010*), which *Garavan et al., 2020* note used 'A very similar task, with the same Go stimulus design features'.

We suggested three preliminary computational models (slowed go processing, guessing, and confusion) that may be able to capture the violation of independence that arise from Design Issue 1. Our preliminary simulations suggest that mean SSRTs are contaminated and SSRT individual differences may be contaminated because of these violations. Existing models that assume independence between the go and stop process (*Logan et al., 2014*; *Logan and Cowan, 1984*) cannot capture the behavior in the ABCD dataset. Additionally, we do not believe that other preliminary stopping models that aim to capture context dependence (*Bissett et al., 2021*) will generalize to the ABCD dataset, given that the violations in ABCD have unique behavioral costs (see *Figure 3*) that are driven by Issue 1. We are pursuing ongoing work to validate the three models above, and we hope the proposed models may be a useful starting point for other modeling efforts in the literature.

Only a subset of the entire ABCD data have been acquired. We have offered prospective suggestions for experimental design changes that we hope will be considered and implemented before future ABCD data is acquired. We recognize that implementing these changes will result in a lack of continuity between how the stop task is implemented across the scans for each subject, but we believe that these issues are significant enough (especially Issues 1–3) that resolving them outweigh the benefit of task continuity. Additionally, some issues involve learning contingencies embedded within the experiment (e.g., Issue 5), so they could exacerbate as subjects have repeated exposures to the task over this longitudinal study. We have also provided retrospective suggestions to data users to contextualize any results that come from these existing data. These suggestions aim to help make the most of the ABCD stopping data, both past and future.

## Materials and methods

In order to complete our analyses, we used the following packages with the following Research Resource Identifiers: Pandas RRID:SCR_018214, NumPy RRID:SCR_008633, Seaborn RRID:SCR_018132, MatPlotLib RRID:SCR_008624, SciPy RRID:SCR_008058, SymPy RRID:SCR_018417, Jupyter Notebook RRID:SCR_018315, scikit-learn RRID:SCR_002577. Some basic methods are included in the Results section. Analysis and simulation code can be found at http://doi.org/10.5281/zenodo.4458428 and http://doi.org/10.5281/zenodo.4458767. Additionally, we lay out the methodology for our modeling and simulations below.

### Modeling and simulations

We proposed three frameworks that may explain the context dependence in the ABCD data. The Independent Race Model (*Logan et al., 2014*; *Logan and Cowan, 1984*) was used as a baseline model for comparison, and it was modified to produce three alternative models which instantiated context dependence: (1) slowed go processing: the go drift rate is reduced by shorter go stimulus presentations at short SSDs, (2) guessing: a propensity to guess at shorter go stimulus presentations at short SSDs, and (3) confusion: both the go and stop drift rates are modulated by shorter go stimulus presentations at short SSDs. A maximum RT threshold of 3000 ms was used to match the response window of the ABCD task.

We modeled slowed drift rates at shorter SSDs with the following formula at SSDs 0–550 ms:

$$\mu_{SSD} = \max\left(\frac{\log\left(\frac{SSD}{550}\right)}{4} + 1, 0\right)$$

where μ is the drift rate. This formula was hand-tuned to have the following characteristics. It results in a 0 drift rate for SSDs of 0 ms (as there is no exogenous go stimulus) and requires at least 10 ms of go stimulus presentation to reach a positive value (though SSDs in ABCD are all a factor of 50 ms). For SSDs greater than 10 ms, the go drift rate rapidly approaches a normal rate with longer go stimulus presentations. This function was applied to modulate the go drift rate in the slowed go processing model and both go and stop drift rates in the confusion model.

To model guessing, a multistep process was applied. First, recognizing that stop-failure RTs for trials with an SSD of 0 ms must be the result of guesses given that participants did not receive any

go stimulus presentation on which to base their decisions (and evidenced by the fact that choice accuracy at that SSD was at chance levels), an ex-Gaussian distribution was fit to those RTs to generate a guessing distribution, and a sampling function was built. We resampled whenever a sample was negative.

Following this, a probability of guessing for trials at each SSD was computed. To do so, the following simplifying assumptions were made: stop-failure RTs at SSDs greater than 0 ms are the result of mixing the go RT distribution with the guessing distribution outlined in the preceding paragraph, no-stop-signal trials do not include guesses, and the choice accuracy of stop-failure trials for a given SSD is the result of this mixing. Therefore, the probability of guessing at a given SSD was found by solving the following formula:

$$ACC_{SSD} = P_{guess|SSD} * ACC_{SSD=0} + 1(1 - P_{guess|SSD}) * ACC_{go}$$

At each SSD, P proportion of simulated trials were sampled from the guess distribution, and 1-P proportion of simulated trials were taken from the no-stop-signal RT distribution.

This simulation paradigm was created to investigate the effects of the ABCD stop task design issues on both mean SSRT estimates and individual differences. To provide all simulation details at once, the investigation of individual differences is described first. To create hypothetical subjects and investigate the effects of the design issues on individual differences, the mean go RT and SSRT were computed for every ABCD subject, and normal distributions were fit to each. In addition, SSD distributions were collected from subjects by subsetting to stop trials with an SSD between 0 and 500, inclusive, and getting the proportion of stop trials within the subset at each SSD. These metrics were not collected for subjects if they experienced Issue 3, or if their probability of responding given a stop signal (P(respond|signal)) was equal to 0 or 1, which indicates degenerate performance, leaving 8,207 subjects. Normal distributions were fit to the go RT and SSRT distributions in order to retrieve the mean and variance. However, because the SSRT values were contaminated by the issues described above, a range of hypothetical variances for the SSRT were tested in order to get a sense for how the between-subject SSRT distribution interacted with the design issues described above to affect individual differences.

The following simulations occurred across four levels of between-subject SSRT variance (85 ms, 25 ms, 5 ms, and 0 ms). These values were compared to the 20 simple stopping datasets in a recent large-scale stopping study (*Bissett et al., 2021*) by computing an SSRT with replacement for each subject in the 20 conditions and finding the standard deviation of SSRTs across subjects within each condition. Negative SSD trials were excluded as they are not present in the ABCD task, and conditions using the Mechanical Turk data were subset to the same subjects who passed the qualitative checks applied in *Bissett et al., 2021*. The 20 conditions had a mean between-subject SD of SSRT = 43 ms with a range of 28 ms–85 ms.

As discussed in the main text, we do not have a trustworthy estimate of individual-subject SSRT, and therefore, we do not have a trustworthy estimate of between-subject variability in SSRT. We chose these values to capture a range that varied from values that largely preserved SSRT as an individual difference metric (SD = 85 ms) down to the limit of no variability such that any preservation of individual differences is driven by differences in go RT and SSD. We simulated 8,207 theoretical individuals. For each simulated individual, a theoretical go RT and SSRT were independently sampled from the normal distributions described above, with resampling until both values were greater than 60 ms (i.e., at least 10 ms greater than the models' nondecision time of 50 ms). These were paired with a unique subject's SSD distribution and converted into drift rates ($\mu$) using the following formula:

$$\mu = threshold/(\{ssrt \text{ or } rt\} - nondecision \text{ } time)$$

where the threshold was set to 100 and the nondecision time to 50 ms.

Each theoretical subject's drift rates were then inserted as parameters to the four competing models, and two sets of simulations occurred. First, a fixed-SSD approach was applied, with 2,500 trials being simulated for each SSD in the range of 0 ms–500 ms, with 50 ms steps, inclusive. Second, a traditional one up one down tracking algorithm (*Levitt, 1971*) was applied, with 25,000 stop trials being simulated. In the staircase approach, the initial SSD was 50 ms to match the initial SSD in the

ABCD dataset, but the maximum SSD was 500 ms, which is different from the max of 900 ms in the ABCD dataset. In both cases, 5,000 go trials were simulated.

Using the fixed-SSD simulations, SSRT was computed at each SSD between 0 and 500 for each subject and each generating model, using the integration with replacement method, following the recommendations of *Verbruggen et al., 2019*. In addition, SSRT was recomputed at each SSD while using each alternative model to generate different underlying distributions for the go process on stop trials (this must be done at the SSD level because the alternative models predict different underlying distributions at different SSDs). For the two models with slowed drift rates, 5,000 go RTs were generated for each SSD. For the guessing model, the original go RTs were augmented with a number of sampled guess RTs such that the proportion of guesses to non-guesses matched that found in the second formula. For example, if the proportion of guesses at a given SSD was found to be 75%, then 15,000 guess RTs would be sampled to combine with the original 5,000 go RTs, creating an RT distribution of which 75% were guesses. These methods allowed us to compare SSRT estimates across different generating models while always making the assumption of context independence.

First, SSRT was computed on the whole of each simulated individual's fixed-SSD data. Second, for each simulated subjects, we sampled an SSD distribution from a real ABCD subject without replacement, and we estimated a simulated mean SSRT by calculating a weighted sum of SSRTs across SSDs:

$$SSRT = \sum_{SSD=0}^{SSD_{max}} P_{SSD} * SSRT_{SSD}$$

Third, an SSRT was computed on the individual's tracking-based simulation data. Rank correlations between alternative models and the standard independent race model using the same SSRT method were compared to describe the minimum and mean rank correlations within the text.

To investigate the effects of the ABCD stop task's design issues on mean SSRT estimates (see *Figure 5*), the above simulations were repeated while setting both the SSRT and go RT between-subject variance to 0 and assigning the mean of both values to the 8,207 simulated subjects. Because each of the 8,207 simulated subjects had the same parameters, this is equivalent to having a single simulated dataset with a large number of trials. For each subject, three SSRT estimates were again computed (weighted, fixed, tracking) per underlying model; these were then averaged across subjects. The means from the tracking-based SSRTs are reported in the manuscript, but the fixed SSD and weighted SSRTs are similar and are presented in the results notebooks linked in the code below.

The code for running these simulations and displaying the simulation results can be found at http://doi.org/10.5281/zenodo.4458767.

## Acknowledgements

We would like to thank Sage Hahn, Hugh Garavan, and their team for identifying an error in a previous version of our manuscript and code that resulted in an inflation in our stop-failure RT estimates. We would also like to thank Stephen J Hanson, Paul Jaffe, Luke Stoeckel, and Jason Zamorano for helpful comments on an earlier draft.

## Additional information

### Funding

No external funding was received for this work.

### Author contributions

Patrick G Bissett, Conceptualization, Data curation, Software, Formal analysis, Supervision, Funding acquisition, Validation, Visualization, Methodology, Writing - original draft, Project administration, Writing - review and editing; McKenzie P Hagen, Henry M Jones, Conceptualization, Data curation, Software, Formal analysis, Validation, Visualization, Methodology, Writing - review and editing;

Russell A Poldrack, Conceptualization, Resources, Software, Formal analysis, Supervision, Funding acquisition, Validation, Visualization, Methodology, Project administration, Writing - review and editing

**Author ORCIDs**
Patrick G Bissett (ID) https://orcid.org/0000-0003-0854-9404

**Decision letter and Author response**
Decision letter https://doi.org/10.7554/eLife.60185.sa1
Author response https://doi.org/10.7554/eLife.60185.sa2

## Additional files

### Supplementary files
• Transparent reporting form

### Data availability

The ABCD dataset is openly available through the NIH Data Archive (https://nda.nih.gov/abcd). Analysis code is available at: http://doi.org/10.5281/zenodo.4458428 and http://doi.org/10.5281/zenodo.4458767.

The following previously published dataset was used:

| Author(s) | Year | Dataset title | Dataset URL | Database and Identifier |
|---|---|---|---|---|
| Casey BJ, Cannonier T, Conley MI, Cohen AO, Barch DM, Heitzeg MM, Soules ME, Teslovich T, Dellarco DV, Garavan H, Orr CA, Wager TD, Banich MT, Speer NK, Sutherland MT, Riedel MC, Dick AS, Bjork JM, Thomas KM, Chaarani B, Mejia MH, Hagler DJ, Cornejo Ramirez MD, Sicat CS, Harms MP, Dosenbach NUF, Rosenberg M, Earl E, Bartsch H, Watts R, Polimeni JR, Kuperman JM, Fair DA | 2018 | Adolescent Brain Cognitive Development [ABCD] | https://nda.nih.gov/edit_collection.html?id=2573 | NIMH Data Archive Collection, 2573 |

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
