## [Decision Letter]

**Acceptance summary:**

The Adolescent Brain Cognitive Development (ABCD) study is an unprecedented longitudinal neuroimaging sample that tracks the brain development of over 10,000 9-10 year olds through adolescence. Three tasks are completed repeatedly in the MRI scanner including the stop-signal task (SST). By analyzing the data of the SST, the authors identified eight design issues that could potentially limit the value of the ABCD. In this paper, prospective solutions for future users next to retrospective solutions for ongoing data users are provided overcoming potential limitations of the ABCD.

**Decision letter after peer review:**

Thank you for submitting your article "Design issues and solutions for stop-signal data from the Adolescent Brain Cognitive Development [ABCD] study" for consideration by *eLife*. Your article has been reviewed by three peer reviewers, and the evaluation has been overseen by Birte Forstmann as the Reviewing Editor and Richard Ivry as the Senior Editor. The following individuals involved in review of your submission have agreed to reveal their identity: Hans Colonius (Reviewer #1); Andrew Heathcote (Reviewer #2); René Huster (Reviewer #3).

The reviewers have discussed the reviews with one another and the Reviewing Editor has drafted this decision to help you prepare a revised submission.

Summary:

This paper focuses on one of the benchmark magnetic resonance imaging (MRI) datasets, the so-called Adolescent Brain Cognitive Development (ABCD). In total, eight design issues observed in the stop signal task of the longitudinal ABCD study by Casey et al. (2018) are pointed out. The design issues are described in detail, ordered by importance, and a number of suggestions are given on how to overcome potential limitations. Given the importance and prominence of the ABCD study in the field of cognitive neurosciences, both the reviewers and editors believe this paper to highlight essential issues in a constructive way. Finally, we believe this paper will elicit a fruitful discussion including the adjustments of the design of the stop signal task.

Overall, this manuscript is well written, interesting, timely and will help resolve the debate in the field. We have the following suggestions to improve the manuscript.

Essential revisions:

1) As the authors suggest, the most important issue is the potential violation of the context invariance assumption due to the variability of the go stimulus duration across different stop signal delays (SSDs). This is a plausible concern even if the number of "clear" violations is relatively small (447 out of 7231 subjects). Nevertheless, the authors' point would be made even more convincing if they could point to some (simulation?) results showing the effect of a weaker go signal at short SSDs on the estimate of the stop signal response time (SSRT).

2) We suggest using the term "context invariance" instead of "context independence" , in order not to confound the assumptions of “context” and “stochastic” independence in the Logan-Cowan race model. It should be pointed out that the prediction of the race model concerning faster stop failures than go responses is conditional on both context invariance AND stochastic independence between go and stop signal processing being true (see Colonius and Diederich, 2018).

3) We recommend you perform and additional analysis: Let us suppose, as you suggest, that the RT distribution of responses to the go signal is indeed affected by the duration of the go signal. As a first approximation, let us assume that the observed RT distribution is a binary mixture of responses: slow RTs to a weak/short go stimulus and fast RTs to a strong/long gos stimulus. Without making specific assumptions about the two components of the mixture, one could employ a mixture distribution test first suggested by Falmagne (1968, British J. Math. Statist. Psychology): The RT ("density") distributions, plotted separately for each SSD and go signal trials, should all cross at one and the same point in time. Of course, this is not a foolproof test but if some evidence in favor of this prediction is found it would strengthen the authors' point.

4) There was some concern on whether the paper is appropriate for *eLife* given our reading of the most relevant aim, which is to publish "studies that use computational methods, models and software to provide important biological insights in all areas of the life sciences". The present paper does not make the sort of positive contribution that this statement seems to imply. Although the paper mentions that "new models for stopping must be developed to accommodate context dependence (Bissett et al., 2019), the latter of which we consider to be of utmost importance to advancing the stop-signal literature", it does not discuss such models and neither does it show the potentially severe consequences of context independence violations in the ABCD data set. Efforts on this issue would strengthen the contribution.

5) The authors write: "Given the above, if analyzing or disseminating existing ABCD stopping data, we would recommend caution in drawing any strong conclusions from the stopping data, and any results should be clearly presented with the limitation that the task design encourages context dependence and therefore stopping behavior (e.g., SSRT) and neuroimaging contrasts may be contaminated". We feel that this recommendation is too lenient and would suggest the following alternative: Unless the ABCD community conclusively shows that the design flaw does not distort conclusions based on SSRT estimates (or any other stop-signal measure), researchers should not use the ABCD data set to estimate SSRTs at all.

6) The authors suggest removing subjects who have severe violations as evidenced by mean stop-failure RT > mean no-stop-signal RT. We are concerned that this recommendation impacts on the representativeness of the sample. Also, this recommendation ignores the fact that violations are not an all-or-none phenomenon but are a matter of degree and can come in varying shapes and sizes.

7) The authors recommend that "any results be verified when only longer SSDs are used, perhaps only SSDs > 200ms". Figure 3 does not seem to support the recommended cut-off of 200ms: at 200ms accuracy is still far from asymptotic.

8) In general, we feel that recommendations based on removing participants and trials are not sufficient such practices will affect the representativeness of the sample and will increase estimation uncertainty and hence decrease power. The real solution here seems to be to develop measurement models that can account for the dependence of the go and the stop process.

---

## [Author Response]

Essential revisions:1) As the authors suggest, the most important issue is the potential violation of the context invariance assumption due to the variability of the go stimulus duration across different stop signal delays (SSDs). This is a plausible concern even if the number of "clear" violations is relatively small (447 out of 7231 subjects). Nevertheless, the authors' point would be made even more convincing if they could point to some (simulation?) results showing the effect of a weaker go signal at short SSDs on the estimate of the stop signal response time (SSRT).

We agree with the reviewers that adding simulations that instantiate a weaker go signal at short SSDs to evaluate the effect on SSRT would strengthen the manuscript. For this reason, we have added a significant new section that presents three new theoretical frameworks that may explain the context dependence in the ABCD data. With these new simulations, we present evidence that mean SSRTs likely are contaminated and individual differences may be contaminated in the ABCD dataset. Please see new section

“Potential Mechanisms Underlying Issue 1”.

We would also like to note that in multiple places we are clear that these models are preliminary. For example, we say:

“For these reasons, they are promising, though preliminary, mechanistic models to explain the violations of context dependence in the ABCD data. Fully establishing these models will require substantial additional work, including parameter fitting and recovery, model recovery, and model comparison.”

Additionally, we have included additional details of the modeling and simulations in the Materials and methods section. Finally, we share open-source code to instantiate all of our modeling and simulation http://doi.org/10.5281/zenodo.4458767. We hope that you agree that this addresses comment 1 and strengthens the manuscript while still remaining cautious about the preliminary nature of these models and simulations.

2) We suggest using the term "context invariance" instead of "context independence" , in order not to confound the assumptions of “context” and “stochastic” independence in the Logan-Cowan race model. It should be pointed out that the prediction of the race model concerning faster stop failures than go responses is conditional on both context invariance AND stochastic independence between go and stop signal processing being true (see Colonius and Diederich, 2018).

We agree with the reviewers that the different forms of independence can be a source of confusion and we have taken multiple steps to address this confusion. We now explicitly distinguish context independence from stochastic independence by saying “it [The Independence Race Model] also assumes stochastic independence which means that the finishing times of the go and the stop processes are independent on any given trial. In this manuscript we focus on the assumption of context independence.”. Later, we also say “The primary way to evaluate context independence is to compare reaction times on go trials to reaction times on stop-failure trials (but note that the prediction of the race model concerning faster stop-failure than go responses is conditioned on both context independence and stochastic independence, Colonius and Diederich, 2018).”

We believe that this addressed the substance of this comment. However, we are not in favor of changing the term “context independence” to “context invariance”. We believe that context independence is the more common terminology (e.g., Verbruggen and Logan, 2009, Neuroscience and Biobehavioral Reviews), and we have used context independence in other recent related work (Bissett, Jones, Poldrack, and Logan, in press) and would like to retain consistency with that.

3) We recommend you perform and additional analysis: Let us suppose, as you suggest, that the RT distribution of responses to the go signal is indeed affected by the duration of the go signal. As a first approximation, let us assume that the observed RT distribution is a binary mixture of responses: slow RTs to a weak/short go stimulus and fast RTs to a strong/long gos stimulus. Without making specific assumptions about the two components of the mixture, one could employ a mixture distribution test first suggested by Falmagne (1968, British J. Math. Statist. Psychology): The RT ("density") distributions, plotted separately for each SSD and go signal trials, should all cross at one and the same point in time. Of course, this is not a foolproof test but if some evidence in favor of this prediction is found it would strengthen the authors' point.

We would like to thank the reviewers for this very important point. We read Falmagne (1968) with great interest and agree that it is an elegant procedure for evaluating whether two or more distributions are made up of a binary mixture of two constant distributions functions.

However, we believe that this procedure is not appropriate for our specific case. Though we agree that the go process on stop-failure trials may arise from a mixture of two processes, for example between fast guesses and stimulus driven responses in our guessing model, the observed stop-failure RT distribution at a given SSD is not a simple realization of this mixture. The observed stop-failure RT distribution at any given SSD is also censored by the stop process, such that predominantly fast responses escape inhibition and populate the observed stop-failure RT distribution and predominantly slow responses are inhibited and therefore are not reflected in the observed stop-failure RT distribution. To be concrete, if at the SSD of 50ms the go process on stop trials is a mixture of 80% fast guesses and 20% slower stimulus driven responses, but only a small number of stop trials at 50ms are stop-failures, then the stop-failure RT distribution may be composed entirely from the fast guesses (and therefore would suggest that the underlying go process at 50ms was not a mixture), and therefore would fail the Falmagne (1968) mixture distribution test, even though the true underlying go process is made up of a mixture of 80% fast guesses and 20% slower stimulus driven responses. Perhaps put more simply, our reading of Falmagne (1968) is that it does not provide a definitive test of binary mixtures if those mixtures are censored by a third process (in this case, the stop process).

We would nonetheless like to thank the reviewer for raising this point, as it provided inspiration for our simulation approach that instantiates (among other models) mixture distributions as the underlying go process on stop trials. Please see Response 1.

4) There was some concern on whether the paper is appropriate for eLife given our reading of the most relevant aim, which is to publish "studies that use computational methods, models and software to provide important biological insights in all areas of the life sciences". The present paper does not make the sort of positive contribution that this statement seems to imply. Although the paper mentions that "new models for stopping must be developed to accommodate context dependence (Bissett et al., 2019), the latter of which we consider to be of utmost importance to advancing the stop-signal literature", it does not discuss such models and neither does it show the potentially severe consequences of context independence violations in the ABCD data set. Efforts on this issue would strengthen the contribution.

Please see Response 1. We have suggested three new models that may explain the mechanisms underlying the violations of context independence in the ABCD data. We show that simulations from each model contaminate mean SSRT and may contaminate SSRT individual differences.

However, we see this work as only a preliminary initial step towards new model building. We have not started model fitting, parameter recovery, and various other steps that would be necessary to provide strong testing of a new model. Therefore, we are careful to couch this as a preliminary work. However, we hope that the reviewers agree that this is a positive computational contribution to the manuscript.

5) The authors write: "Given the above, if analyzing or disseminating existing ABCD stopping data, we would recommend caution in drawing any strong conclusions from the stopping data, and any results should be clearly presented with the limitation that the task design encourages context dependence and therefore stopping behavior (e.g., SSRT) and neuroimaging contrasts may be contaminated". We feel that this recommendation is too lenient and would suggest the following alternative: Unless the ABCD community conclusively shows that the design flaw does not distort conclusions based on SSRT estimates (or any other stop-signal measure), researchers should not use the ABCD data set to estimate SSRTs at all.

Particularly in light of our new simulations (see Response 1), we agree with the reviewers that our previous recommendation was too lenient. We have adjusted the language to closely align with the suggestions from the reviewers:

“Given the above empirical evidence and simulations, and as suggested by our reviewers, unless the ABCD community shows that this design issue does not distort conclusions based upon SSRT estimates (or any other stop-signal measure), researchers should not use the ABCD dataset to estimate SSRTs, and should use the neuroimaging data with caution”

6) The authors suggest removing subjects who have severe violations as evidenced by mean stop-failure RT > mean no-stop-signal RT. We are concerned that this recommendation impacts on the representativeness of the sample. Also, this recommendation ignores the fact that violations are not an all-or-none phenomenon but are a matter of degree and can come in varying shapes and sizes.

We agree with the reviewers and have adjusted the manuscript along the lines that they suggest. We begin this paragraph by saying:

“We also suggest two practical suggestions to attempt to avoid some of the most extreme violations, but we note significant shortcoming of these solutions.”

We go on to say:

“However, this may impact the representativeness of the sample. Also, though this may eliminate some of the subjects who most severely violated, violations are not an all-or-none phenomenon and this will likely leave in subjects who violate, albeit to a less extreme degree. Additionally, our above simulations demonstrate that violations of context independence can manifest as go slowing (as in slowed go drift rate) or go speeding (as in fast guesses at short SSDs), the latter of which would not be captured by this criterion at all. Therefore, this first suggestion should not be taken as sufficient to confidently eliminate the influence of violations.”

7) The authors recommend that "any results be verified when only longer SSDs are used, perhaps only SSDs > 200ms". Figure 3 does not seem to support the recommended cut-off of 200ms: at 200ms accuracy is still far from asymptotic.

We agree with the reviewers that a cutoff that removes SSDs <= 200ms may be too inclusive. However, if our criterion is that choice accuracy would need to fully asymptote then essentially all of the stop data would be removed. We attempted to balance the desire for caution with the desire to provide positive recommendations for how a data user might be able to make some use of these flawed data. In light of the above considerations, we have added the following:

“Additionally, Figure 3 shows that choice accuracy on stop failure trials does not asymptote at 200ms, so a longer cutoff (and additional data loss) would more confidently avoid severe violations.”

8) In general, we feel that recommendations based on removing participants and trials are not sufficient such practices will affect the representativeness of the sample and will increase estimation uncertainty and hence decrease power. The real solution here seems to be to develop measurement models that can account for the dependence of the go and the stop process.

We agree with the reviewers and we have adjusted our recommendations in multiple ways. As discussed in response 6, we add caveats to the approach of removing subjects. As discussed in response 7, we add caveats to removing only short SSDs trials. We also adjusted the following sentence from “should” to “may”:

“Neither of these two suggestions resolve Issue 1 but they may eliminate the subjects and trials most likely to show severe violations, respectively”.

We also end that section in the following way:

“In order to resolve Issue 1, the different go durations that encourage context dependence must be eliminated (see prospective suggestions), or new models for stopping must be developed to accommodate context dependence, the latter of which we consider to be of utmost importance to advancing the stop-signal literature. We attempt to make preliminary progress in this direction with the models that we instantiate in the above simulations and in companion work that suggests preliminary theoretical frameworks that can account for context dependence (Bissett et al., 2021). However, these are very preliminary proofs of concept and much more work is necessary to fully flesh out a viable model for stopping that can accommodate context dependence. Additionally, we believe that Design Issue 1 introduced idiosyncratic violations (i.e., reduced choice error rates) that will require idiosyncratic computational solutions.”